# Thermal Bed Design for Temperature-Controlled DNA Amplification Using Optoelectronic Sensors

**DOI:** 10.3390/s24217050

**Published:** 2024-10-31

**Authors:** Guillermo Garcia-Torales, Hector Hugo Torres-Ortega, Ruben Estrada-Marmolejo, Anuar B. Beltran-Gonzalez, Marija Strojnik

**Affiliations:** 1Department of Electronics, University Center for Exact Sciences and Engineering, University of Guadalajara, Av. Revolucion 1500, Guadalajara 44840, Jalisco, Mexico; hector.hhto@gmail.com (H.H.T.-O.); ruben.estrada@academicos.udg.mx (R.E.-M.); anuar.beltran@academicos.udg.mx (A.B.B.-G.); 2Optical Research Center, Leon de los Aldamas 37150, Guanajuato, Mexico; mstrojnik@gmail.com

**Keywords:** LAMP, DNA amplification, microfluidics, thermal control, optoelectronic sensors

## Abstract

Loop-Mediated Isothermal Loop-Mediated Isothermal Amplification (LAMP) is a widely used technique for nucleic acid amplification due to its high specificity, sensitivity, and rapid results. Advances in microfluidic lab-on-chip (LOC) technology have enabled the integration of LAMP into miniaturized devices, known as μ-LAMP, which require precise thermal control for optimal DNA amplification. This paper introduces a novel thermal bed design using PCB copper traces and FR−4 dielectric materials, providing a reliable, modular, and repairable heating platform. The system achieves accurate and stable temperature control, which is critical for μ-LAMP applications, with temperature deviations within ±1.0 °C. The thermal bed’s performance is validated through finite element method (FEM) simulations, showing uniform temperature distribution and a rapid thermal response of 2.5 s to reach the target temperature. These results highlight the system’s potential for applications such as disease diagnostics, biological safety, and forensic analysis, where precision and reliability are paramount.

## 1. Introduction

Advances in molecular diagnostics have underscored the need for rapid and accurate nucleic acid amplification techniques. Loop-Mediated Isothermal Amplification (LAMP) is a powerful method known for its high specificity and sensitivity in DNA amplification under isothermal conditions [1,2]. The integration of LAMP into microfluidic lab-on-chip (LOC) systems, termed μ-LAMP, allows for miniaturization and automation, greatly enhancing the efficiency of molecular diagnostics However, the success of μ-LAMP is highly dependent on precise thermal control systems that can maintain stable temperatures to ensure optimal DNA amplification [3].

This paper presents a novel thermal bed design based on printed circuit board (PCB) copper traces with FR−4 dielectric materials, offering precise and stable temperature control essential for DNA amplification. The thermal bed consists of a spiral-shaped copper trace covering an area of approximately one square inch. By utilizing a MOSFET in a TO-252 package and driven by a PWM signal from a dsPIC microcontroller, the system dynamically controls the current flowing through the trace, which acts as a resistive heating element. The bed integrates a temperature sensor to provide real-time feedback for a PID-controlled loop, ensuring the target temperature is maintained with minimal fluctuation [4,5]. The optoelectronic sensors further monitor colorimetric changes, essential for tracking the amplification progress. This modular and repairable design is particularly suitable for applications in disease diagnosis, biological safety monitoring, and forensic analysis [6,7,8,9,10,11,12].

One of the key advantages of this thermal bed design is its simplicity and cost-effectiveness due to the use of PCB manufacturing techniques. The system’s flexibility in terms of configuration makes it highly adaptable, enabling potential scalability to various applications that require precise thermal control. For instance, its ability to interface with sensors and respond dynamically to environmental changes allows for a high degree of reproducibility in experiments. Additionally, the modular nature of the design makes it suitable for a range of similar applications, where control over thermal cycling and parameter characterization based on sensor data is critical. This adaptability offers significant advantages in fields requiring scalable, reproducible, and flexible solutions for temperature management.

The μ-LAMP method, widely utilized for nucleic acid amplification, facilitates DNA synthesis in approximately 20 min with high specificity, sensitivity, and minimal initial fluid requirements. Lab-on-chip (LOC) technologies have potential for numerous applications, including disease diagnosis, biological safety, food analysis, and pathogen detection. The successful implementation of this technology could revolutionize sample processing in forensics, significantly reducing the time required to handle backlogged cases, such as rape kits. The integration of LAMP with microfluidic chip technology has led to the development of μ-LAMP [13,14,15,16,17,18,19,20,21,22].

There is growing interest in implementing biological processes for chemical evaluation and analysis across various disciplines. Biology, chemistry, and medicine increasingly demand technologies that can perform laboratory procedures quickly and with minimal manual intervention. These advances often employ autonomous self-regulating devices to reduce human error and improve precision [23,24,25,26,27,28,29].

Optimizing parameters to improve DNA amplification reliability involves selecting appropriate thermoelectric components, designing the thermal fluid chamber, and integrating optical sensors with instrumentation and control systems. These elements are critical in monitoring and characterizing DNA amplification processes that involve feedback control mechanisms. Efficient temperature management is vital for the functionality of the μ-LAMP process, underscoring the importance of robust thermal control systems [30,31,32,33,34,35,36,37,38,39,40].

## 2. Methodology

The analysis focuses on the temperature increase caused by Joule’s law, which is generated by the flow of current in a closed circuit. Precise control of the electric current is achieved through a PWM-regulated MOSFET controlled by a dsPIC microcontroller running a Proportional–Integral–Derivative (PID) algorithm. This setup ensures that the current flow is modulated according to the feedback from a temperature sensor, which continuously monitors the thermal bed’s temperature. The MOSFET, operating in a switching mode, allows fine-tuned adjustments to the current by varying the duty cycle of the PWM signal. This method minimizes power loss and ensures efficient heat generation. The choice of materials, such as the copper trace on the PCB, and the physical parameters of the thermal bed were selected to optimize thermal response and maintain uniform temperature distribution.

The device includes all design phases, from introducing the fluid mixture to completing the DNA amplification process. Two optoelectronic sensors play a key role in process control: one for temperature measurement and another for color detection, monitoring the amplification progress. The temperature is sampled at a single reference point to correlate with the sample’s temperature, verified through repeated simulations with varying system parameters. Detecting the precise completion of the process, especially with the miniaturized motility sensor, presents a challenge. An optical sensor detects color changes due to turbidity shifts in the sample, calibrated to ensure accurate control of the μ-LAMP process with miniaturized electro-optical sensors and precise temperature measurements. Introducing a small volume of fluid samples for DNA testing poses a technological challenge not covered in this work.

This paper presents a model for the electric current distribution leading to controlled temperature increases, crucial for a DNA amplification device operating at 62 °C. We introduce a novel thermal bed design that utilizes advances in semiconductor technology, specifying the thermoelectric parameters of copper tracing for effective implementation. Using the finite element method (FEM), we simulate the copper trace as a thermal bed for the sample chamber and evaluate its temperature performance under various electrical current levels.

### 2.1. Thermal Bed Design

Figure 1 illustrates the block diagram outlining the architecture of the DNA amplification device. The hardware consists of a copper trace integrated into a printed circuit board (PCB), which is heated by controlling the current flow through pulse width modulation (PWM). This thermoelectric system generates the necessary heat to raise and maintain the temperature within the sample chamber, ensuring the conditions required for isothermal amplification. The control subsystem, implemented via a Proportional–Integral–Derivative (PID) algorithm running on a dsPIC microcontroller, ensures that the temperature remains within the predefined limits. A temperature sensor, specifically the TMP110 I2C module, is strategically placed within the fluid chamber to provide real-time feedback for the temperature control system. This digital sensor is directly connected to the microprocessor, allowing for accurate and continuous monitoring of the chamber’s thermal conditions. The TMP110 comes precalibrated from the manufacturer, ensuring reliable temperature readings without the need for additional calibration steps during the experimental setup. This sensor’s precision, with a tolerance of ±1.0 °C, is well suited for the LAMP process, where maintaining a stable temperature is critical for ensuring high specificity and sensitivity in DNA amplification. Small deviations in temperature can affect the reaction efficiency, making precise thermal control essential to avoid incomplete or erroneous amplification. The sensor’s data are used to adjust the PWM signal in real time, ensuring the current delivered to the heating element is modulated appropriately to maintain the desired temperature.

To monitor the progress of DNA amplification, a modified optical turbidity sensor is employed as an optical color sensor, detecting colorimetric changes that signal the completion of the reaction. The microprocessor interprets the output of the sensor by continuously measuring the RGB values of the sample. A threshold-based algorithm processes these RGB values, where a predefined change—such as a 40% increase in the green channel intensity—signals the end of the amplification. The color shift, typically from transparent to green, is compared with reference data from known samples to validate the completion of the process. Additionally, a time-based control ensures that the system halts after a maximum duration of 30 min, should the color change not reach the expected threshold. This dual verification system of real-time color monitoring and time control ensures the accuracy and reliability of the DNA amplification process.

In our system design, hardware elements are shown with orange boxes; communication protocols with blue lines and boxes; and sensors, detectors, the PC interface, and the power system for the PCB trace are represented with green boxes. A dsPIC^®^ DSC microprocessor controls the device, while the power supply in the upper-right corner distributes energy to the thermoelectric heating stage via PWM. An RGB LED, digitally controlled by the microprocessor, provides process visualization. Data transmission is illustrated in the lower-left part, showing compatibility with Bluetooth, ZigBee, and other wireless or wired communication protocols.

We developed a mathematical model to evaluate the current distribution across the PCB traces, optimizing the physical, mechanical, and geometric parameters for chip implementation. Thermoelectric simulations using the finite element method (FEM) confirmed that the proposed temperature increase per current is appropriate. The PCB design software also simulated the copper trace configuration to ensure a uniform thermal distribution throughout the fluid volume.

After simulations, the selection of electronic components for the test PCB was performed to validate the design. The thermal bed uses a specific current, adjusted by the PWM, to meet the electrical specifications of the components, including transistors and controllers, ensuring both functionality and longevity. The DNA amplification process, accelerated by the μ-LAMP procedure, operates at a stable temperature of 62 °C. The methodologies described can also model and optimize other processes requiring different temperatures, such as those starting around 100 °C.

Providing a controlled temperature environment is crucial for processes in fields such as biology, medicine, chemistry, and forensics [14,35]. The primary goal is to replicate DNA or develop new methods for analyzing biological events. Various heating mechanisms have been used to control the temperature in these systems. Recent advances focus on thermal management in μ-LAMP devices, including silicon arrays for temperature control in PCR amplification [31,32], micro-coils for temperature regulation in lab-on-chip platforms, and thermistors and thin-film technologies for PCR applications [34,35,41,42].

This study follows the IPC Industrial and Thermoelectric Standards Model (Association of Connecting Electronic Industries [43]). After selecting the physical characteristics of the copper trace and power supply, simulations are conducted to analyze the thermal resistance of the PCB, establishing the relationship between the temperature increase and the required electrical current. The current and its control unit automatically regulate the temperature rise without manual intervention.

To optimize DNA amplification, specific thermal requirements must be met. The control mechanisms raise the temperature to the desired operating point and maintain it for the duration needed for DNA amplification. The mechanical parameters of the PCB trace are then defined.

Figure 2 illustrates the characteristics of the PCB used in the thermal bed design. Figure 2a shows the cross-section of an analyzed PCB segment. The copper thickness is consistent across the PCB layer, corresponding to the selected copper foil: 1 Oz ${\mathrm{ft}}^{-2}$. The width of the copper trace is defined in the design program and measured in mils (1 mil = 0.001 in = 0.025 mm). Adjusting the trace width changes its impedance, allowing control over the current capacity for that segment. The target temperature increase and electrical current are considered to propose the optimal trace width.

The design and manufacturing standard for PCBs, IPC-2221A ((D-31b) 199846), provides an approximation for the temperature increase in a trace as a function of the electric current flowing through it. This expression limits the physical geometry of the trace width relative to the amount of current, given by
(1)$$I=\gamma \Delta T_{C}^{1.2}A^{0.75}.$$

Here, *I* is the electrical current in amperes [A], *A* is the transverse cross-sectional area in [mm^2^], ΔT is the temperature increase in [K], and γ is a constant trace layer of copper such that: $\gamma_{1}$ = 0.048, $\gamma_{2}$ = 8.024. Here, $\gamma_{1}$ is used for the external layer of the PCB and $\gamma_{2}$ for its internal layer, known as stripline and microstrip, respectively. This approach works well for panel PCBs with a dielectric thickness of 0.8 mm of FR−4 material and copper conductors with a width of 4.252 mil (0.108 mm) or greater.

Once we have selected the power supply, we can deduce other system parameters from it. Specifically, a copper trace of 10-mil (0.254 mm) width and a thickness of 1 Oz ft^2^ (a standard PCB specification, equal to 34.79μm) of copper is considered. Then, the current may be limited to less than 1 A. According to the first law of thermodynamics, the total energy of a closed system is conserved. The only way the power of a system changes is when it crosses the boundaries of the system. This law further determines the way energy crosses system boundaries [44]. The following expression is valid for a closed system:(2)$$\Delta P_{st}^{tot}=Q-W.$$

Here, ΔP is the total change in the power stored in the system in [W]; $\frac{dQ}{dt}$ is the net heat transferred to the system per unit time, also in [W]; and $\frac{dW}{dt}$ is the net amount of work generated by the system per unit time. The derivative of the first law of thermodynamics may also be applied to a controlled volume or even an open system. This analysis is important for determining the unknown temperature, as in our trace segment. Using it, we choose the current necessary for the requisite temperature increase, considering that no external (mechanical) power sources or sinks exist that might provoke a temperature change. The parameter obtained through this analysis is the temperature change rate as a function of current. The first law of thermodynamics and its derivative with time apply at any instant; therefore, the following expression is valid.
(3)$$P_{st}=P_{in}+P_{g}-P_{out}.$$

Here, $P_{st}$ in [W] is the rate of increase of energy within a volume in [W], $P_{in}$ is the rate of input energy transfer into the volume in [W], $P_{out}$ is the rate of energy transfer out of a volume, and $P_{g}$ is the rate of energy generation, likewise in [W]. All of this takes place according to the geometry and materials of the components. Using Equation (Equation 3), we determine the current necessary to increase the temperature, considering that no external mechanical power sources could generate a temperature change. Equation (Equation 3) may be interpreted to mean that the rate of increase in thermal or mechanical energy stored in a volume must be equal to the rate at which thermal and mechanical energy enter the volume minus the rate at which thermal and mechanical energy leave the volume, plus the rate at which all energy (thermal and other) is generated within the volume:(4)$$P_{st}=\frac{d\rho VcT}{dt}.$$

Figure 3a illustrates how the energy of the system changes with time for a short copper segment, representing a defined volume, and how it is interconnected. The electric current *I* controls the amount of thermal energy generated per unit of time inside the volume using the resistive heating mechanism. Assuming that no energy flows into the short segment of the copper trace, we obtain the following expression from Equation (Equation 3):(5)$$P_{st}=P_{g}-P_{out}.$$

The resistive losses produce the rate of energy dissipation, then
(6)$$P_{g}=I^{2}R_{e}L.$$

Here, *I* is the electrical current in amperes [A], $R_{e}$ is the electrical resistance of the copper trace per unit of length [Ω$m^{-1}$], and *L* is the length of the trace segment in [mm]. The temperature increment of the copper trace is uniform within the volume because a concise segment is considered. This temperature increase may be found as a function of the energy increment rate within the copper trace volume:(7)Pg=qV.
where *q* is the rate of energy increment per unit of volume. Substituting the volume parameters for the geometry in our design, we have
(8)$$q=\frac{i^{2}R_{e}}{\left(w)(th\right)}.$$

Here, *w* and th are the width and thickness of the copper trace, in [m] respectively. In this system, the thermal energy includes convection and radiative transfers away from the surface of the copper trace. The next equation represents the convection and radiative losses, considering only the surfaces exposed to air and the background temperature [44].

Where *h* is the heat transfer coefficient for convection in [W $m^{-2}$
$K^{-1}$]; σ is the Stefan–Boltzmann constant, $5.67\times 10^{-8}$ [W $m^{-2}$
$K^{-4}$]; ϵ is the emissivity of copper that is between 0.03 and 0.04, without units; and $T_{sur}$ and $T_{\infty }$ are the temperatures of the outer layer of the copper trace volume and of the background in [K], respectively. The rate of energy increase within a volume is given by
(9)$$P_{out}=h\left(\left(w+th\right)L\right)\left(T-T_{\infty }\right)+\epsilon \sigma \left(\left(w+th\right)L\right)\left(T^{4}-T_{sur}^{4}\right).$$

Here, ρ is the mass density in [kg $m^{-3}$], c is the specific heat capacity in [J k$g^{-1}$
$K^{-1}$], and *V* is the volume of the copper trace, [m^3^]. In the case of the copper trace, the mass density is 8933 kg $m^{-3}$, and the specific heat capacity is 385 [J k$g^{-1}$
$K^{-1}$]. Substituting Equations (4) through (8) into Equation (Equation 9), we solve it for the copper trace’s temperature increase as a current function:(10)$$\frac{dT}{dt}=\frac{i^{2}R_{e}h\left(\left(w+th\right)L\right)\left(T-T_{\infty }\right)+\epsilon \sigma \left(\left(w+th\right)L\right)\left(T^{4}-T_{sur}^{4}\right)}{d\rho VcT}.$$

We summarize the quantities for the sake of completeness. *T* is the temperature increase in [K]; *t* is the time in [s]; *I* is the current in amperes [A]; and $R_{e}$ is the electrical resistance per unit length [Ω
$m^{-1}$], which depends on the copper trace’s geometry and must be calculated using its width (*w*), thickness (th), and length (*L*). These parameters are design specific and should be measured from the PCB or obtained from design files. The convection heat transfer coefficient *h* in [W $m^{-2}$
$K^{-1}$] varies with cooling conditions and should be experimentally determined, though typical values for air range between 5 and 25 W/m²·K. $T_{sur}$ (surface temperature) and $T_{\infty }$ (ambient temperature) are measured directly, with $T_{sur}$ monitored by a TMP110 sensor integrated into the system. Standard values for other parameters are ϵ (emissivity of copper) between 0.03 and 0.04, σ (Stefan–Boltzmann constant) as $5.67\times 10^{-8}$ W $m^{-2}$
$K^{-4}$, ρ (density of copper) at 8933 kg $m^{-3}$, and *c* (specific heat) at 385 J k$g^{-1}$
$K^{-1}$. These do not require experimental determination.

Equation (Equation 5) may be evaluated numerically. Our system is ten mils wide and 1 Oz ${\mathrm{ft}}^{-2}$ thick copper trace (as proposed in the design). We obtain the following numerical estimate for the increase in temperature as a current function for the proposed copper tracing using a numerical evaluation presented graphically in Figure 3b. We see that any final temperature in the range between room temperature (25 °C) to above 100 °C can be reached with a current of less than 0.5 A through the trace. Thus, many processes may be achieved with the technique described here.

We note that a current smaller than 1 A is required to raise the temperature to the process temperature of 62 °C. To increase the system responsivity, the possibility of working with 1 to 2 W power supplies is also explored. Employing a power supply any larger than 1 W may damage the device itself or its constituent electronic components: the temperature increase may exceed their recommended operating conditions. When using a power source with power significantly smaller than 1 W, the system may take too long to reach, or it may never reach, the temperature required for the amplification process.

### 2.2. Finite Element Method (FEM) Simulations

A section of the copper trace was drawn in the ANSYS™ program. This simulation program works on the theory of finite elements for structures. Its dynamic mesh is presented in Figure 4a. To ensure that the mesh size did not significantly impact the results, a grid dependence test was conducted. The test involved refining the mesh and observing the changes in the calculated temperature at critical points. As shown in Figure 5, the temperature changes became negligible as the mesh size was refined, indicating that the solution had become independent of the mesh size. This ensured that the mesh was sufficiently fine for accurate results without unnecessary computational overhead.

The simulation was performed for an electric current input through a copper trace to determine the current density distribution and the resulting temperature evolution through space and time. First, the current density inside the trace volume was calculated. The current density is a vector quantity whose magnitude is the electric current per unit area to the current flow’s direction. Figure 4b shows the vector current density as a function of the location on the trace. The current density is related to the electric current as follows:(11)I=j∫dS.

Again, *I* is the current in amperes, [A]; *j* is the current density in [A$m^{-2}$]; and *S* is the surface area in [m^2^]. In our simulations, copper was considered the preferred material for selecting the trace. This material is also commonly used in the manufacturing of PCBs.

Figure 6 shows two PCB trace schemes using the finite element method. Figure 6a shows the trace with the initial parameters applicable to the system under design. The preliminary current of 500 mA is proposed for an initial temperature of 25 °C. In addition, an electrical reference potential and a fixed reference point for anchoring the motion are considered, shown in the lower right corner. Figure 6b shows the distribution of the electric current density vectors throughout the system. It is observed that the magnitude of the current density is greatest when the trace changes direction by 90 degrees. Arrows represent the distribution of the electric current density vectors, their direction, and magnitude.

After calculating the electric current density vectors and using knowledge of the intrinsic parameters of the implemented trace material, it is finally possible to determine the temperature increase in the copper trace as a function of time using the finite element method [45,46,47].

The calculated temperature distributions on the copper trace are shown for two specific times in Figure 7a,b. We can appreciate how the trace starts at a reference temperature of 25 °C in Figure 7a. The first half of the trace has already started to be heated at time t = 0.5 s, while the end segment remains at room temperature. After a specific, relatively short time at t = 2.5 s, the trace reaches the target temperature at least for the first half of the trace length, which in this case is 62 °C.

The simulations of the thermoelectric stage of the device are an iterative process, leading to ever-better design solutions compatible with the device’s requirements. With the final design, we are confident that the physical characteristics and thermal response of the unit are compatible with the device performance requirements. During the final, optimized design and simulation iteration, we determine that the trace configuration of 10 mil (0.254 mm) width and 1 Oz ${\mathrm{ft}}^{-2}$ (34.79 μm) height is the optimal one for our prototype development.

### 2.3. Optoelectronic Sensor Integration

The geometric information of the parameters obtained with the thermal simulations is the data used in a PCB design program. The preliminary design resulting from the upper and lower layers of the thermal bed is shown in Figure 8a,b. The thermal stage of the PCB has an effective area of 39.7 × 28.45 mm^2^. The thermal traces satisfy the design rule of 10 mils (0.254 mm) wide and 10 mils (0.254 mm) of spacing between the traces.

The central hole in the thermal bed is designed to accommodate an external light source if necessary. The four holes near the corners of the card guide the positioning, alignment, and assembly of the fluid chamber, which is constructed from multiple layers of acrylic sheeting placed on top of the card. The green pads on the right side are for attaching a connector. If a jumper cable shorts the connector, the current flows only through the top part. If the connector is open, the current flows through both parts, allowing us to adjust the system’s rate of temperature increase by changing the connector’s state.

Technological advances in processors and reconfigurable systems allow for the precise implementation of control algorithms, such as those used to stabilize system temperatures. Proportional–Integral–Derivative (PID) controllers, both in their classical and modified forms, are widely utilized due to their reliability, adaptability, and efficiency in maintaining stable reference temperatures [48,49,50]. In this study, a PID controller is employed to regulate the temperature of the system. The PID controller functions as a feedback system, where the proportional ($K_{p}$), integral ($K_{i}$), and derivative ($K_{d}$) components work together to minimize the error between the setpoint and the measured temperature.

The proportional term ($K_{p}$) directly impacts the system by adjusting the control output based on the current temperature error. This immediate correction helps the system quickly respond to deviations from the reference temperature. However, high proportional gains can lead to overshooting as observed in the system’s response. The integral term ($K_{i}$) accumulates the error over time, ensuring that any steady-state error is eliminated, although excessive integral action may introduce oscillations. The derivative term ($K_{d}$) anticipates future trends by considering the rate of change of the error, reducing overshoot and improving system stability, particularly during rapid changes in temperature.

This control system is programmed through the configuration of the dsPIC microprocessor registers and libraries. Sensor readings are processed, converted, and adjusted for microprocessor input, while the output from the PID controller is scaled to modulate the PWM duty cycle, which controls the current delivered to the system. The control remains active from the start of the process until DNA amplification is completed, at which point the control signals deactivate.

To ensure accurate modeling of the thermal response, a transfer function is derived based on the electrical properties of the PCB copper trace, which acts as the system’s heating element. The trace exhibits resistive, inductive, and capacitive properties that influence the system’s dynamic behavior. Specifically, the resistance R=0.024Ω, inductance L=0.2μH, and capacitance C=1 pF define the RLC circuit that models the trace. This transfer function characterizes the relationship between the PWM control signal and the temperature rise, representing the PCB dynamic response. The copper trace’s resistance determines the current flow, which, in combination with the inductance and capacitance, shapes the system’s second-order response. The saturation element ensures that the PWM duty cycle is limited to a maximum of 100% by the microprocessor’s programming. The transfer function for this model is expressed as
(12)$$H\left(s\right)=\frac{1}{\left(0.2\times 10^{-6}\right)\left(1\times 10^{-12}\right)s^{2}+\left(0.024\right)\left(1\times 10^{-12}\right)s+1}$$

Figure 9 presents the block diagram of the second-order control system, illustrating the closed-loop feedback cycle and the PID parameters. The system continuously compares the measured temperature with a reference of 62 °C, adjusting the control signal accordingly. For different temperature setpoints, only minor adjustments to the programming are necessary.

Figure 10 demonstrates the system’s response to a step input of 62 °C. In Figure 10a, the blue line represents the input signal, while the red line shows the temperature response. The system exhibits a typical second-order response with overshooting and minor oscillations. The PCB serves as the plant, where the temperature increase is directly proportional to the current regulated by the PWM signal. The control signal, which scales the PWM signal to match the required temperature increase, is shown in Figure 10b. The initial peak (overshoot) results from the proportional term’s dominant action at the start of the process, as the system compensates for the large initial temperature difference with an amplified control signal. Over time, the PID controller stabilizes the temperature as the proportional, integral, and derivative actions converge. This overshooting behavior is typical in second-order systems, and the inherent filtering effect of the electronic components mitigates the oscillations, functioning similarly to a low-pass filter.

The measured values by the sensors are processed and converted to acceptable values as input to the microprocessor. The output value of the control system is scaled, providing a multiplier factor for the register that controls the duty cycle of the PWM. This system is active from the start of the process until the amplification system is completed when the cycle deactivates the output commands.

Figure 11 presents a photo of the assembled prototype. Figure 11a shows part of the device that incorporates the control module with a microprocessor to manage sensors and power generation. This sub-assembly controls power delivery to the thermal stage and receives data from the temperature and color sensors. The microprocessor on the left side of the board meets the minimum performance requirements for these control functions.

The dsPIC33FJ128, a 16-bit midrange microprocessor, is shown with connectors at the bottom for ISP programming and top connectors for serial communication, allowing direct connection to a computer or wireless module. Figure 11a also illustrates the power control stage, with an external power source connectable in the center. The microprocessor interfaces with the controller to manage the gate current of the Infineon IRLB8721 MOSFET, chosen for its low gate threshold voltage ($V_{\mathrm{GS}(\mathrm{th})}$) of 1.0 V to 2.35 V, making it ideal for a 3.3 V PWM signal. Its low on-state resistance ($R_{\mathrm{DS}(\mathrm{on})}$) of 8 mΩ ensures efficient operation with minimal power loss, while its high current-handling capacity (62 A) supports rapid thermal changes in the PCB resistive element. Alternative MOSFETs, such as the STL9N3LLH6 (1.0 V to 1.8 V $V_{\mathrm{GS}(\mathrm{th})}$, 22 mΩ$R_{\mathrm{DS}(\mathrm{on})}$) or the CSD18563Q5A (6.5 mΩ$R_{\mathrm{DS}(\mathrm{on})}$, 80 A), also provide viable options for controlling current flow. By modulating the gate with a PWM signal, the system adjusts the current through the copper trace to generate and control the heat needed for precise temperature regulation.

On the right side of Figure 11a, the color and temperature sensors connected to the microprocessor are visible. Only this right portion of the design is shown in Figure 11a, as the full schematic is beyond this publication’s scope. The illumination source, an RGB LED, is matched with an RGB color sensor, and includes a temperature sensor to measure the temperature difference between the control base and the fluid layer. This setup helps estimate the temperature and heating time constant of the fluid chamber when heating is applied only to the base. Figure 11b displays the RGB lighting source module for controlling and processing the color and temperature sensors.

Figure 12 shows the PCB layouts for the control assemblies depicted in Figure 11a,b. The design layout highlights the elegance and neatness of the fabricated subsystem, mounted on the prototype’s upper section. The I2C communication protocol connectors are used for sensors and the RGB LED, with one for each color and a shared white connector.

## 3. Results

The performance of the thermal bed in maintaining precise temperature control is evaluated both experimentally and through finite element method (FEM) simulations. The target temperature for DNA amplification in the μ−LAMP process is set at 62 °C.

The thermal bed is tested with a real-time temperature monitoring setup using integrated optoelectronic sensors. Figure 13a shows that the thermal bed rapidly reaches the target temperature of 62 °C within 2.5 s when a current of 0.5 A is applied. The measured temperature shows minor oscillations around the setpoint, indicating the control system’s effectiveness in maintaining a stable thermal environment. These oscillations are within ±0.5 °C of the target temperature, demonstrating the precision of the temperature control system.

FEM simulations are conducted to predict the temperature distribution along the copper trace and validate the uniformity of the heating. Figure 13a illustrates the simulated temperature distribution at different intervals. The first half of the trace reaches the target temperature quickly, while the other half gradually heats up, confirming the expected thermal behavior of the system. The results indicate that the temperature distribution is highly uniform along the trace, with a deviation of less than 2% throughout the length, even in a steady state.

The effectiveness of the thermal bed in facilitating DNA amplification is evaluated by monitoring the fluorescence intensity over time, which correlates with the DNA concentration during the amplification process. As depicted in Figure 13b, the fluorescence intensity follows a typical sigmoid curve, which reflects the different phases of the amplification process. Initially, during the lag phase, there is minimal increase in fluorescence intensity as the reaction components adjust to the conditions and prepare for DNA replication. This phase is followed by the exponential phase, where a sharp increase in fluorescence between 10 and 20 min indicates rapid DNA replication, with the thermal bed successfully maintaining optimal conditions for efficient amplification. Finally, the plateau phase is reached around the 25 min mark, where the fluorescence intensity stabilizes as the amplification process slows and eventually nears completion. These distinct phases further demonstrate the thermal bed’s ability to support the dynamic nature of the amplification process, ensuring both rapid and stable DNA replication.

The optoelectronic sensors integrated into the thermal bed play a crucial role in achieving precise temperature control and real-time monitoring of the DNA amplification process. The temperature sensor is calibrated to ensure accurate readings within the temperature range required for the μ-LAMP process. The calibration is verified through multiple trials, demonstrating a high correlation $\left(R^{*}>0.99\right)$ between the sensor readings and reference temperatures, confirming the sensor’s reliability for real-time temperature monitoring. The colorimetric sensor, used to detect changes in the optical properties of the reaction mixture, is validated against known color standards. The sensor shows high sensitivity and specificity, detecting tiny color changes associated with DNA amplification. This capability is crucial to determine the endpoint of the amplification process without manual intervention.

The design efficiency of the thermal bed is evaluated based on its thermal response time, energy consumption, and compatibility with microfluidic systems. The thermal bed demonstrates a rapid thermal response, reaching the target temperature in less than 3 s. The thermal stability is maintained over extended periods, with minimal drift, essential for prolonged DNA amplification procedures. The design currently relies on passive thermal dissipation through the PCB for cooling. However, the system architecture allows for the future integration of active cooling mechanisms, such as heat sinks or Peltier cells, to accelerate the cooling process when necessary. This flexibility makes the design adaptable for more demanding thermal cycling protocols, where rapid temperature transitions are required. The power consumption of the thermal bed is measured to be less than 1 W during steady-state operation, making it highly energy efficient. This low power requirement is particularly advantageous for portable and battery-operated microfluidic devices, ensuring prolonged use without frequent battery replacements or recharges.

The results demonstrate the thermal bed’s suitability for integration into various microfluidic and lab-on-chip devices. Its ability to maintain precise temperature control and facilitate rapid DNA amplification has significant implications: the thermal bed can be used in point-of-care diagnostics for rapid pathogen detection, enhancing the speed and accuracy of disease diagnosis in clinical settings. The device also has the capability to detect and amplify DNA sequences, which can be applied to monitor biological safety in various environments, including food safety testing and environmental monitoring. The rapid and reliable DNA amplification facilitated by the thermal bed can help reduce processing times in forensic applications, particularly in analyzing crime scene samples and backlogged forensic cases.

DNA Amplification Progress: This graph represents the progress of DNA amplification over time, measured through the fluorescence intensity. The amplification follows a typical sigmoid curve, indicating successful DNA replication as the process progresses, which is detected by the increase in fluorescence intensity.

These results demonstrate the ability of the LOC device to maintain precise temperature control and effectively facilitate DNA amplification.

## 4. Discussion

The novel thermal bed design offers significant advantages over traditional thermal control systems used in DNA amplification. Using PCB copper traces and FR−4 dielectric materials, the thermal bed provides a compact, modular, and repairable platform easily integrated into various microfluidic devices. Optoelectronic sensors enhance the system’s reliability and functionality, allowing for the real-time monitoring and control of the amplification process.

The uniform temperature distribution and precise control achieved by the thermal bed are critical for the success of the μ-LAMP technology. Maintaining a stable thermal environment ensures high specificity and sensitivity in DNA amplification, making the thermal bed suitable for various applications in molecular diagnostics, including disease detection, biological safety monitoring, and forensic analysis.

However, despite its advantages, several potential limitations and challenges need to be addressed to ensure the reliability and performance of the system. One of the key challenges is the calibration of sensors, particularly the temperature and colorimetric sensors. Incorrect calibration can result in inaccurate temperature readings, leading to fluctuations that negatively impact the amplification process by reducing the specificity and sensitivity of the DNA amplification. Similarly, the colorimetric sensor’s sensitivity to small changes in color during the amplification process is critical for detecting the endpoint of the reaction. Poor resolution in the sensor or an incorrect relationship between the expected color change and the sensor’s response can affect the system’s ability to detect subtle variations, compromising the accuracy of the quantification.

Moreover, the selection of appropriate sensors plays a crucial role in the overall performance of the system. Sensors with inadequate precision or limited dynamic range could fail to capture the necessary data, reducing the effectiveness of the feedback control loop. Ensuring that the sensors are properly matched to the system’s requirements is essential for maintaining the tight control necessary for successful DNA amplification.

The design of the control system itself must also be carefully considered. While the current system uses a PWM-controlled MOSFET with a PID feedback loop, factors such as response time and sensor placement can introduce delays or inaccuracies in the control process, affecting the uniformity of the temperature distribution. Future improvements could include exploring more advanced control algorithms or introducing additional mechanisms, such as thermal insulation or heat sinks, to enhance the system’s thermal stability and response time.

Finally, environmental factors, such as electromagnetic interference (EMI) from the MOSFET switching, could also affect sensor accuracy and system performance. Ensuring proper shielding and grounding of the system is crucial to mitigate these risks, especially in sensitive diagnostic applications.

In conclusion, while the novel thermal bed design offers promising advantages in terms of energy efficiency, scalability, and integration into various microfluidic devices, addressing these potential limitations will be essential for maximizing the reliability and performance of the system in practical applications.

## 5. Conclusions

We present a comprehensive approach to designing a novel thermal bed for temperature-controlled DNA amplification using optoelectronic sensors. The proposed thermal bed, constructed with PCB copper traces and FR−4 dielectric materials, offers a precise and reliable temperature control crucial for the μ-LAMP process. The integration of optoelectronic sensors facilitates real-time monitoring and feedback control, significantly enhancing the system’s functionality and ensuring high specificity and sensitivity in DNA amplification. The experimental results and the finite element method (FEM) simulations confirm that the thermal bed maintains a stable thermal environment, which is essential to achieve consistent DNA amplification outcomes. The device’s ability to precisely control temperature in microfluidic settings makes it a versatile tool suitable for various applications in molecular diagnostics, including disease detection, biological safety monitoring, and forensic analysis.

Furthermore, the modular and repairable design of the thermal bed represents a significant advancement in lab-on-chip technologies, providing a versatile platform that can be easily adapted to other temperature-sensitive applications. Future work will focus on optimizing the response times of the thermal bed and exploring its scalability for mass production, potentially expanding its applicability in clinical and research settings. These developments underscore the potential for the integration of advanced thermal control systems and optoelectronic sensors into miniaturized diagnostic devices, paving the way for more efficient, reliable, and automated molecular diagnostics.

## Figures and Tables

**Figure 1 sensors-24-07050-f001:**
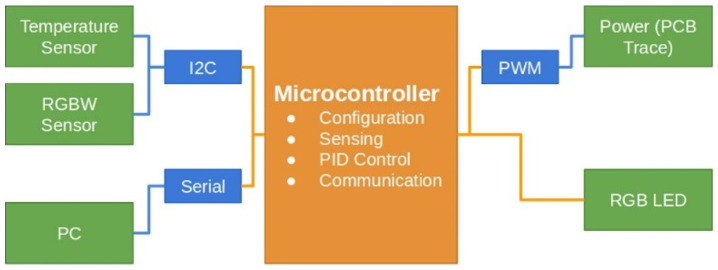
Block diagram illustrating the sensors and hardware interconnections within the lab-on-chip (LOC) system. The diagram shows how a microprocessor coordinates the communication between sensors and a computer using standard communication protocols, enabling the precise control and monitoring of the DNA amplification process.

**Figure 2 sensors-24-07050-f002:**
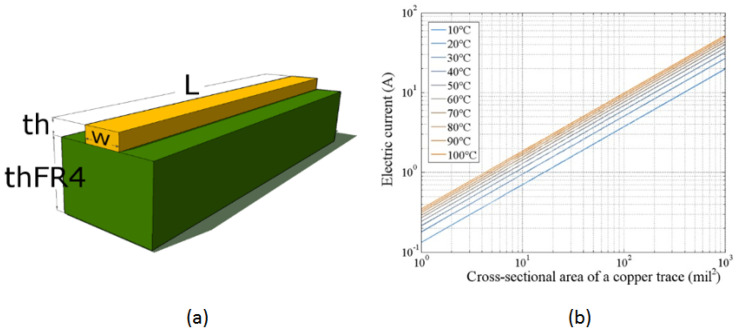
Characteristics of the PCB used in the design of the thermal bed. (**a**) Cross-sectional view of the copper trace on the FR−4 substrate material, illustrating the physical dimensions and layout of the trace. (**b**) Graph showing the relationship between the electric current and the transverse cross-sectional area of the copper trace for various final temperature increments under a 1 W power source. The graph demonstrates temperature increases from 10 °C to 100 °C in 10 °C intervals, highlighting the thermal response characteristics of the copper trace.

**Figure 3 sensors-24-07050-f003:**
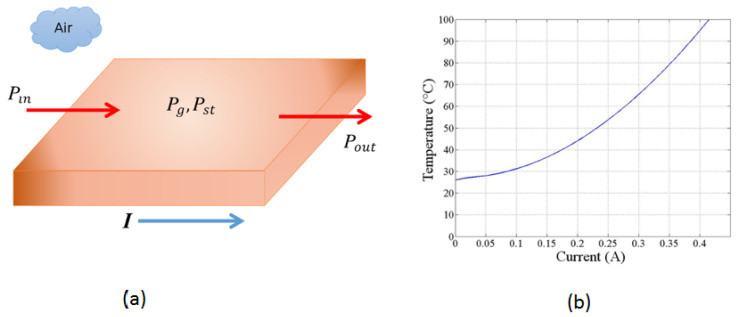
Sketch used to model the thermal energy transport along the copper trace and the calibration curve of temperature as a function of electrical current. (**a**) Geometry used to determine the energy flow balance through a segment of the copper trace at a specific moment in time. The resistive losses generate heat as electrical current flows through the copper trace, resulting in a temperature increase. There is no external thermal input or output energy in this model. (**b**) Temperature as a function of electrical current for the copper trace in the PCB, showing an approximately quadratic relationship.

**Figure 4 sensors-24-07050-f004:**
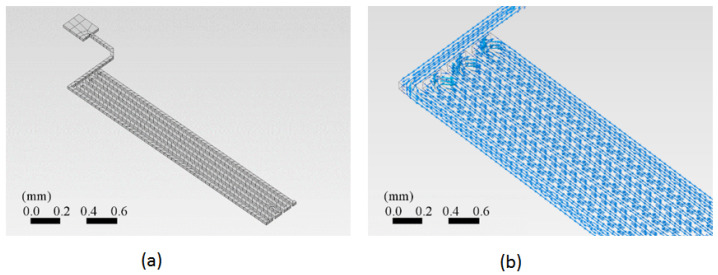
Two schemes of the PCB trace analyzed using the finite element method (FEM). (**a**) The dynamic mesh suitable for FEM, showing the geometric layout of the copper trace. The mesh density increases in areas of greater interest, such as corners, to enhance the accuracy of the analysis. (**b**) The vector current density as a function of location on the trace, indicating how electrical current is distributed along the copper trace.

**Figure 5 sensors-24-07050-f005:**
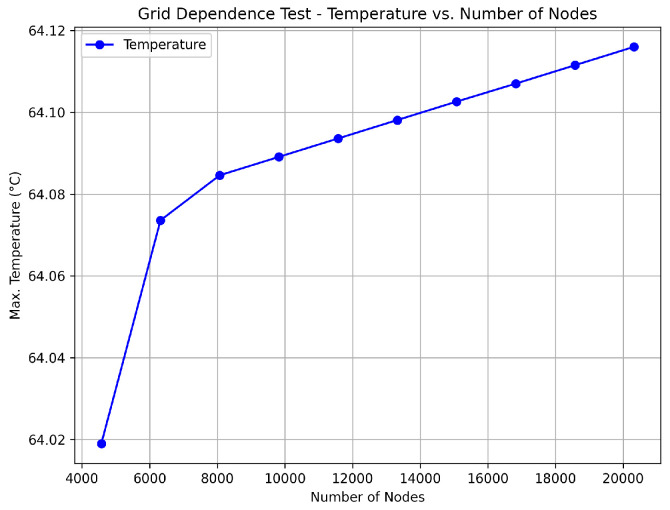
Grid dependence test results showing the temperature evolution as a function of the number of mesh nodes. The results indicate that beyond a certain number of nodes, the temperature stabilizes, confirming that further mesh refinement is unnecessary.

**Figure 6 sensors-24-07050-f006:**
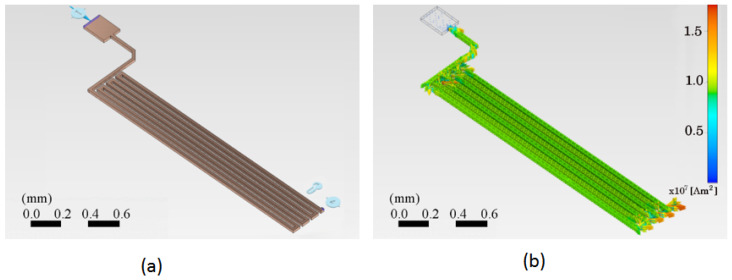
Geometrical and current density parameters of the PCB trace. (**a**) The geometrical configuration of the copper trace shows the reference potential points and motion anchoring positions used in the analysis. (**b**) The electric current density vectors are a function of location on the trace surface. The color scale on the right indicates the current density values, ranging from 0 to $1.7\times 10^{7}$ A $m^{-2}$. The current density magnitude is highest where the trace bends at 90 °C.

**Figure 7 sensors-24-07050-f007:**
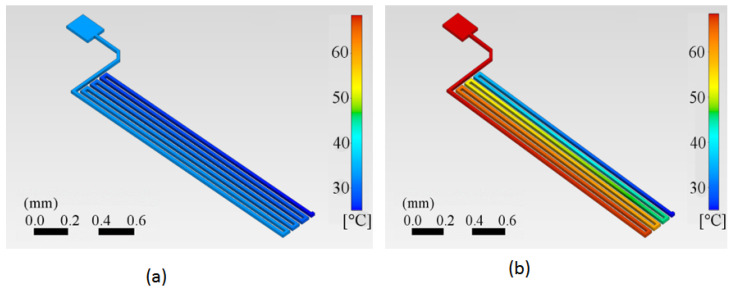
Temperature distribution in the copper trace starting from an initial temperature of 25 °C. (**a**) At 0.5 s after a current of 0.5 A is applied, the first half of the trace has begun to heat, while the end segment remains at room temperature, as indicated by the darker blue color. (**b**) At 2.5 s with the same current of 0.5 A, the first half of the trace has reached the target temperature of 62 °C, while a portion of the trace still remains at room temperature, represented by the blue color.

**Figure 8 sensors-24-07050-f008:**
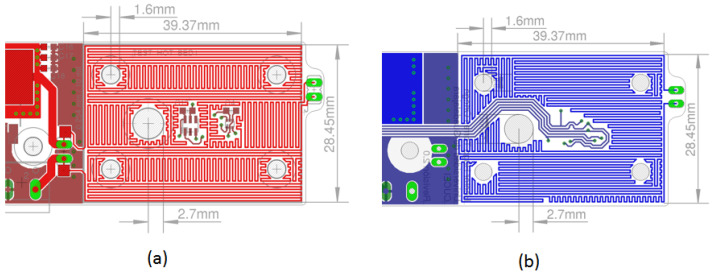
Design of the thermal stage using the EAGLE^TM^ CAD program. (**a**) The layout of the upper layer shows the arrangement of copper traces designed to ensure optimal uniformity when heating the sample chamber. (**b**) The layout of the lower layer illustrates a similar trace arrangement to maintain uniform heating throughout the sample chamber.

**Figure 9 sensors-24-07050-f009:**
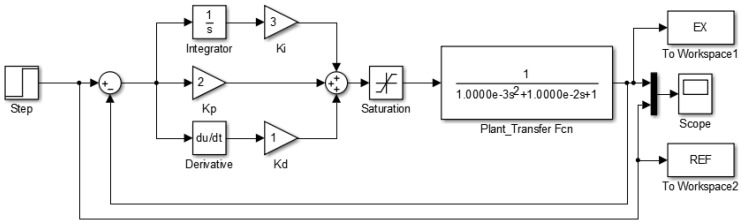
Block diagram of the PID control system implemented in the DNA amplification device. The PCB (plant) is modeled as an RLC circuit to represent the resistive (R), inductive (L), and capacitive (C) components of the thermal control system. This model allows precise regulation of the temperature within the device to ensure optimal conditions for DNA amplification.

**Figure 10 sensors-24-07050-f010:**
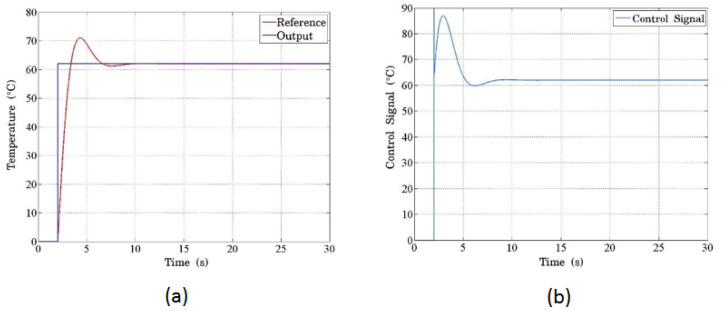
(**a**) The temperature response programmed in the controller using the proposed thermal model. The blue line represents the reference signal, scaled and expressed in temperature units, showing how the system tracks the desired setpoint. (**b**) The simulated control signal for the system demonstrates the initial peak with a saturation limitation due to the sharp change from a high-step input. This peak behavior results from the control system’s attempt to reach the target temperature rapidly.

**Figure 11 sensors-24-07050-f011:**
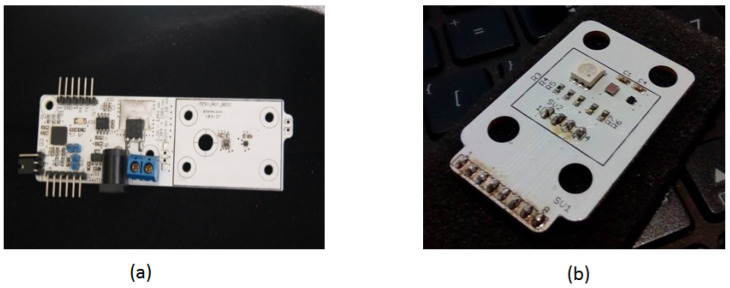
A photo of the assembled control systems for the DNA amplification device. (**a**) The primary PCB control unit, showing the main components responsible for regulating temperature and system operations. (**b**) The secondary control PCB with a protective cover, which houses additional circuitry for managing the device’s sensor inputs and outputs.

**Figure 12 sensors-24-07050-f012:**
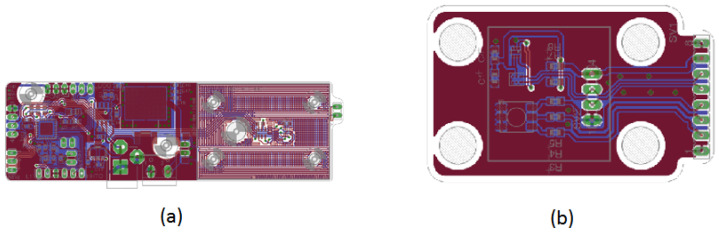
PCB layout design for the DNA amplification device control unit. (**a**) Top view of the primary PCB control layout, showing the placement of components and copper traces for optimal performance. (**b**) Bottom view of the primary PCB control layout, illustrating the routing of connections and additional components to ensure reliable operation of the control system.

**Figure 13 sensors-24-07050-f013:**
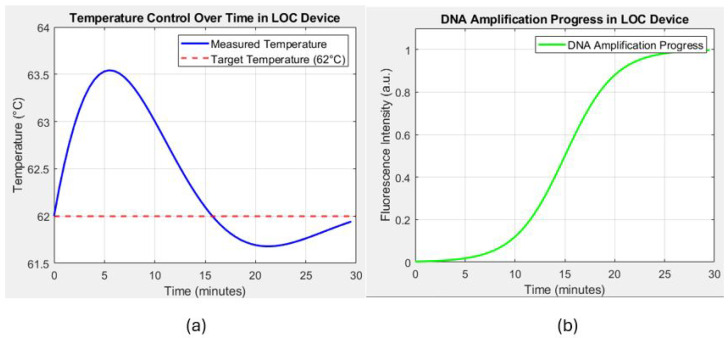
(**a**) Temperature control over time in the lab-on-chip (LOC) device, showing the measured temperature (blue line) and the target temperature (62 °C, red dashed line). The graph demonstrates the ability of the device to maintain a stable temperature around the target setpoint, which is critical for effective DNA amplification. (**b**) Progress in DNA amplification in the LOC device, represented by the fluorescence intensity over time (green line). The sigmoid curve indicates the successful amplification of DNA, with the process nearing completion around the 25 min mark.

## Data Availability

Data are contained within the article.

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
