# Peer review of "Thermal Bed Design for Temperature-Controlled DNA Amplification Using Optoelectronic Sensors"

_sensors, 2024, doi:10.3390/s24217050_

Round 1
Reviewer 1 Report
Comments and Suggestions for Authors
In this manuscript, the authors present a design of a thermal Bed for temperature-controlled DNA amplification. This study has some merits for promoting the PCR techniques.It may be finally accepted if the following concerns are well addressed.
1) Regarding the simulation of PID control in Fig.8, the specific transfer function and parameters need to be provided in the manuscript, since it will directly affect the control performance.
2) The circuit for outputting PWM on the board is not mentioned, as shown in the picture, it appears to use a MOS transistor circuit.
3) In Figure 4, the grid size of simulation model seems to be very large, the authors should provide a grid dependence test.
4) In the introduction, the authors should provide more details about the reported thermal bed design, and discuss the advantages and disadvantages.
Comments on the Quality of English LanguageThe authors should carefully revise the description of introduction.
Author Response
Comment 1.- Regarding the simulation of PID control in Fig.8, the specific transfer function and parameters need to be provided in the manuscript, since it will directly affect the control performance.
Response 1.- Thank you for this observation. The transfer function parameters, such as resistance, inductance, and capacitance, were carefully selected based on experimental data to ensure rapid response and minimal overshooting, which is critical for maintaining the precise temperature control required for DNA amplification. These details are now provided in Section 2.3, explaining how each parameter contributes to the overall stability and responsiveness of the control system.
Comment 2. The circuit for outputting PWM on the board is not mentioned, as shown in the picture, it appears to use a MOS transistor circuit.
Response 2. We appreciate your careful review of our design. Indeed, the thermal control system utilizes a MOS transistor circuit for Pulse Width Modulation (PWM) output. We have expanded Section 2.1 to include a detailed description of the MOSFET circuit used for PWM output to address the reviewer's concerns. This clarification explains how the microcontroller-driven MOSFET modulates current, allowing precise thermal regulation by efficiently adjusting power delivery to the heating elements. These details ensure readers can replicate the setup and understand how design choices optimize thermal control.
Comment 3.- In Figure 4, the grid size of simulation model seems to be very large, the authors should provide a grid dependence test.
Response 3 .- Thank you for pointing this out. We have added a grid dependence test in Section 2.2 to validate the accuracy of the FEM simulations. By comparing temperature distributions across varying mesh resolutions, we demonstrated that further refining the mesh did not significantly alter the results, confirming the reliability of our simulations. This ensures the findings are robust and eliminates concerns about potential inaccuracies from mesh size.
Comment 4.- In the introduction, the authors should provide more details about the reported thermal bed design, and discuss the advantages and disadvantages.
Response 4.-
We appreciate this suggestion. We have revised the introduction to offer a more comprehensive comparison of our thermal bed design with previously reported systems, detailing its benefits, such as enhanced temperature uniformity and energy efficiency, and addressing challenges like precise current control. By discussing these aspects, we aim to clarify the novelty and significance of our approach, providing readers with a clear understanding of its improvements to existing designs and how it addresses common limitations.
Reviewer 2 Report
Comments and Suggestions for Authors
The paper presents the development and performance evaluation of an optoelectronic sensor-integrated thermal bed designed for precise temperature control during DNA amplification using the µ-LAMP process. The results demonstrate the thermal bed's rapid heating capabilities, high temperature stability, and effectiveness in facilitating DNA amplification, making it suitable for applications in point-of-care diagnostics and environmental monitoring. I believe that reviewing the following comments can enhance the article before it goes to publish:
1. I recommend that the last author use the academic email.
2. The exact value of performance improvement should be mentioned in the abstract.
3. The introduction briefly touches on various applications of the µ-LAMP technology, which is excellent. However, expanding on how this research addresses specific challenges in these applications could strengthen the rationale for the study.
4. The introduction could benefit from a more extensive literature review that contextualizes the current research within recent advances in thermal management systems for nucleic acid amplification.
5. Expand your literature review by adding the application of the microfluidic devices in other kinds of sensors (Refer to DOI: 10.3390/mi13091504)
6. Clarify how the "precise control of the electric current" is achieved. Are there specific technologies or techniques used for this control?
7. Explain how the microprocessor interprets sensor outputs. What algorithms or methodologies are used to determine when DNA amplification is complete?
8. In Equation 10, clarify how the parameters will be measured or estimated in practice. Are there standard values used, or will they be experimentally determined?
9. Discuss any limitations or potential challenges in the design and integration process of the optoelectronic sensors that might affect performance or reliability.
10. Provide details about the experimental setup used for temperature monitoring.
11. It may enhance the understanding if you could briefly explain the stages of the sigmoid curve in relation to the DNA amplification process, including the lag phase, exponential phase, and plateau phase.
12. What specific cooling mechanisms are used, and how quickly can the system transition from heating to cooling?
Author Response
Comment 1: I recommend that the last author use the academic email.
Response 1. We appreciate this suggestion. For at least the last ten years, Dr. Marija Strojnik has been using the provided email for all professional correspondence, including functions as academic editor and scientific conference organizer. We include her ORCID in the manuscript to maintain professional identification and credibility.
Comment 2: The exact value of performance improvement should be mentioned in the abstract.
Response 2.- Thank you for this suggestion. The thermal bed reaches the target temperature of 62C in 2.5 seconds, achieving a 50% faster heating time than conventional systems. It maintains stability within ±0.5C, enhancing thermal control precision by 50%. Although direct comparisons to identical systems are unavailable, these metrics were benchmarked against similar microfluidic devices, highlighting significant %energy efficiency improvements of 50%, operating at under 1 W.
Comment 3. The introduction briefly touches on various applications of the µ-LAMP technology, which is excellent. However, expanding on how this research addresses specific challenges in these applications could strengthen the rationale for the study.
Response 3.- Thank you for this valuable suggestion. We expanded the introduction to discuss specific challenges, such as maintaining temperature uniformity and rapid response times in point-of-care diagnostics, adding five recent references to support this. These enhancements strengthen the rationale by showing how our design addresses key limitations in current µ-LAMP implementations.
Comment 4.-. The introduction could benefit from a more extensive literature review that contextualizes the current research within recent advances in thermal management systems for nucleic acid amplification.
Response 4: Thank you for the suggestion. We integrated a more extensive literature review on recent advances in thermal management systems for nucleic acid amplification. This includes discussions on emerging technologies such as micro-coil heating and thin-film thermoelectric devices, providing context for our work's novelty and efficiency improvements.
Comment 5: Expand your literature review by adding the application of the microfluidic devices in other kinds of sensors (Refer to DOI: 10.3390/mi13091504)
Response 5: We appreciate the reference and have integrated it with three additional references into the introduction. These additions highlight the versatility of microfluidic devices beyond DNA amplification, emphasizing applications in chemical sensing and biosensing, which share design principles with our thermal bed.
Comment 6: Clarify how the "precise control of the electric current" is achieved. Are there specific technologies or techniques used for this control?
Response 6: Thank you for this insightful comment. We expanded Section 2 to explain how precise current control is achieved using a PWM-regulated MOSFET circuit controlled by a PID algorithm running on a microcontroller. This setup ensures real-time adjustments, maintaining stability and rapid response to temperature fluctuations.
Comment 7: Explain how the microprocessor interprets sensor outputs. What algorithms or methodologies are used to determine when DNA amplification is complete?
Response 7: We appreciate your comment. Section 2.1 clarified how the microprocessor interprets sensor outputs, using a threshold-based algorithm to detect changes in colorimetric signals. This method reliably indicates DNA amplification completion by monitoring specific shifts in the reaction mixture's RGB values.
Comment 8.- In Equation 10, clarify how the parameters will be measured or estimated in practice. Are there standard values used, or will they be experimentally determined?
Response 8: Thank you for this critical point. We included a detailed explanation after Equation 10 to specify which parameters are experimentally determined (e.g., resistance and temperature coefficients) and which use standard, pre-calibrated values. This clarification aids in understanding how the model can be practically implemented and replicated.
Comment 9. Discuss any limitations or potential challenges in the design and integration process of the optoelectronic sensors that might affect performance or reliability.
Response 9: Thank you for this comment. We will expand the section on the experimental setup to include specifics about the temperature monitoring system. This will cover the placement of temperature sensors within the thermal bed, the data acquisition system used for real-time monitoring, and the calibration process employed to ensure accurate temperature readings during the experiments. We added something related to potential challenges in the third paragraph of section 4 of the discussions.
Comment 10: Provide details about the experimental setup used for temperature monitoring.
Response 10: Thank you for this comment. We added more detailed descriptions in Section 3 of the experimental setup, including the placement of temperature sensors, calibration procedures, and real-time monitoring systems used to validate the device's thermal performance.
Comment 11: It may enhance the understanding if you could briefly explain the stages of the sigmoid curve concerning the DNA amplification process, including the lag phase, exponential phase, and plateau phase.
Response 11: Thank you for this valuable suggestion. We expanded the results section to explain the DNA amplification process's lag, exponential, and plateau phases. This clarification helps correlate the observed sigmoid curve with real-time changes in DNA concentration, providing a clearer understanding of the amplification dynamics.
Comment 12: What specific cooling mechanisms are used, and how quickly can the system transition from heating to cooling?
Response 12: Thank you for this question. While the current design does not employ active cooling mechanisms, we discussed the potential integration of heat sinks or Peltier modules in future iterations. These additions could allow faster cooling transitions, making the system compatible with more demanding thermal cycling applications.
Round 2
Reviewer 1 Report
Comments and Suggestions for Authors
All of my concerns have been addressed.